# Variability of Meat and Carcass Quality from Worldwide Native Chicken Breeds

**DOI:** 10.3390/foods11121700

**Published:** 2022-06-09

**Authors:** Antonio González Ariza, Francisco Javier Navas González, Ander Arando Arbulu, José Manuel León Jurado, Juan Vicente Delgado Bermejo, María Esperanza Camacho Vallejo

**Affiliations:** 1Department of Genetics, Faculty of Veterinary Sciences, University of Córdoba, 14071 Cordoba, Spain; angoarvet@outlook.es (A.G.A.); anderarando@hotmail.com (A.A.A.); juanviagr218@gmail.com (J.V.D.B.); 2Institute of Agricultural Research and Training (IFAPA), 14004 Cordoba, Spain; mariae.camacho@juntadeandalucia.es; 3Agropecuary Provincial Centre, Diputación Provincial de Córdoba, 14071 Cordoba, Spain; jomalejur@yahoo.es

**Keywords:** data mining, biodiversity, genetic resources, local breeds, physical properties characterization, chemical profile

## Abstract

The present research aimed to determine the differential clustering patterns of carcass and meat quality traits in local chicken breeds from around the world and to develop a method to productively characterize minority bird populations. For this, a comprehensive meta-analysis of 91 research documents that dealt with the study of chicken local breeds through the last 20 years was performed. Thirty-nine traits were sorted into the following clusters: weight-related traits, histological properties, pH, color traits, water-holding capacity, texture-related traits, flavor content-related nucleotides, and gross nutrients. Multicollinearity problems reported for pH 72 h post mortem, L* meat 72 h post mortem, a* meat 72 h post mortem, sex, firmness, and chewiness, were thus discarded from further analyses (VIF < 5). Data-mining cross-validation and chi-squared automatic interaction detection (CHAID) decision tree development allowed us to detect similarities across genotypes. Easily collectable trait, such as shear force, muscle fiber diameter, carcass/pieces weight, and pH, presented high explanatory potential of breed variability. Hence, the aforementioned variables must be considered in the experimental methodology of characterization of carcass and meat from native genotypes. This research enables the characterization of local chicken populations to satisfy the needs of specific commercial niches for poultry meat consumers.

## 1. Introduction

Two parallel poultry industries can be distinguished in most developing countries. In this regard, while there is a commercial industry that uses hybrid strains of high-yielding broilers or layers, local farm practices are based on a rather sustainable production through the use of dual-purpose autochthonous breeds with consequent lower yields. The representativity of these two categories varies greatly depending on the country. However, in low-income countries, native breeds may represent up to 90% of the poultry population [1,2].

The main difference between these two production systems lies in the management practices that are carried out. On the one hand, commercial genotypes are typically raised in individual cages or confined in flocks, ranging from 100 to 200 (small) to over 10,000 individuals (large). Moreover, feeding in industrial poultry farms is based on commercially compounded feed, and the facilities are often located close to urban areas. On the other hand, local genotypes are bred in family farms in rural and peri-urban areas, in small flocks of 10–30 birds, which are usually fed with household scrap and smaller amounts of grains and commercial feed [3,4,5].

Even if they may present lower productivity, native breed birds can efficiently reproduce without the need for incubators or artificial breeding, facing the harsh conditions of rural environments and alternative production systems [6,7]. Native breed birds are rather agile, fly, roost in trees, and display enhanced anti-predator strategies [8,9]. Local breeds have been deemed more resistant than commercial broilers and layers to bacterial and parasitic diseases [10,11]. Furthermore, the food products derived from these birds (meat and eggs) are generally preferred to those of commercial lines, not only among rural communities but also in urban areas [12,13].

A widespread concern of the loss of the valuable and irreplaceable heritage represented by poultry genetic material has been reported worldwide during the last decade. The replacement and crossbreeding of native breeds with commercial high-producing breeds is often referred to as the main cause to which to ascribe such a loss [14]. In this way, there is a potential risk that low genetic variability could jeopardize the industry in the event of a serious disease outbreak with new virus strains [2].

Recent studies have shown that observed heterozygosity is higher in native breeds when compared with brown-eggshell and white-eggshell layers and broilers [15]. Thus, genetic diversity may play a pivotal role in the improvement of breeds and adaptation of livestock populations to climate change, emerging diseases, pressure on land and water, and changing market demands. For this, it is important to ensure that animal genetic resources are conserved and used sustainably.

FAO developed a set of Sustainable Development Goals indicators for livestock diversity directly linked to the DAD-IS database. Among the elements that the set comprises, we find the number of plant and animal genetic resources for food and agriculture secured in either medium- or long-term conservation facilities (2.5.1.) and the proportion of local breeds, classified as being at risk of extinction (2.5.2.). According to the Domestic Animal Diversity Information System (DAD-IS) FAO database [16], 57.73% of chicken breeds are classified to hold an unknown risk status, 13.17% are endangered, 9.99% hold a critical status, 6.08% are extinguished, and only 9.18% are not at risk. The DAD-IS tool acts as an information resource, which reports the average number of gaps in ex situ collections of selected crop gene pools and quantifies the classification of local breeds [17].

Meat is one of the most nutritious foods and is considered to be an essential element of diet by a large percentage of the world’s population to reach an optimal growth and development in humans. Among all animal-derived foods, poultry meat contains a large number of desirable characteristics, including low lipid content and high concentrations of polyunsaturated fatty acids (essential in human nutrition) [18]. Furthermore, chicken meat is an extensively acknowledged, essential-amino-acids-rich source of quality and cheap protein, which presents important minerals and vitamins (such as threonine, lysine, methionine, cysteine, and tryptophan) [19]. Poultry meat quality has been reported to be highly influenced by bird genetics [20]. Moreover, several husbandry factors, including feeding, breeding, and management (pre-slaughter, stunning, slaughter and post-slaughter procedures, chilling, and storage conditions) can somehow influence carcass and meat traits [18,21].

From all the different features used to define carcass and meat quality, the most traditionally frequently used are weight, pH, and water-holding capacity (WHC) traits [22]. However, during the last 20 years, the trends have changed toward the development of carcass technology and study methods, allowing us to deepen the knowledge of other traits, such as texture, content of flavor-related nucleotides, chemical composition, histological properties, and muscle color characterization. For this reason, recent research has focused on characterizing the aforementioned quality traits and their applicability in chicken meat production [23,24,25,26,27]. Such studies not only verse on the characterization of the quality of the meat and carcass from native breeds but also promote the definition of such traits as breeding criteria, which in turn improves the efficiency of conservation and breeding programs based on the profitable sustainability of their products.

Therefore, the present article first aims to determine the differential clustering patterns that carcass- and meat-quality-related traits describe in worldwide local chicken breeds. Second, the benefits that derive from the use of data mining are verified through the development of a functional tool to quantify the similarities and dissimilarities across those native breeds for which product quality or quantity analyses have been developed. The outcomes of the present study will enhance the conservation and sustainability opportunities of the breeding program of autochthonous chicken breeds. Furthermore, the tool developed in the present study may help define the samples considered when planning future research involving minority populations of birds in terms of breed selection and control group or parameters definition when seeking the evaluation of particular meat- and carcass-quality-related parameters.

## 2. Material and Methods

### 2.1. Data Collection

Data collection was carried out as previously described by González Ariza et al. [28], McLean and Navas Gonzalez [29], and Iglesias Pastrana et al. [30]. For this, the repositories at www.google.scholar.es and www.sciencedirect.com (accessed on 21 March 2022) were used [31]. While the aforementioned platforms include tools that enable data extraction for analysis, other repositories, such as www.ncbi.nlm.gov/pubmed/ (accessed on 28 April 2022), do not, which prompted their exclusion as information sources. Non ‘open-access’ documents were accessed through the library service of the University of Córdoba (Córdoba, Spain).

The following keywords were used: carcass or meat quality/characterization. Each one followed with the words: local/native/indigenous/autochthonous poultry or chicken breed (or any related term in their semantic fields [28,32]). Data collection comprehensively contains documents published from 1968 to 2021. Document search was finished in January 2022 to ensure that all publications of the year 2021 had been considered on the whole.

Only documents involving breeds, which appeared in the DAD-IS catalog as native breeds were considered in the statistical analyses [33]. Out of all the documents selected, the 91 research documents approaching the evaluation of carcass or meat pieces of native chicken breeds described in the clusters referred to in Table 1 were retained. All the publications searched were in English and Spanish languages. In addition, the age at slaughter and sex of the individual from whom the information was collected were considered. The sex groups (sex and reproductive status) considered were male, female, both (if males and females were used in the papers without differentiation), capon and poulard.

As the information reported in the literature may make use of different units to quantify the carcass and meat quality, the corresponding conversion was carried out so that the results across breeds could be comparable. All units were converted to the most frequently used units across documents (Table 1).

Following this methodology, the selected documents were recorded in a database. Observations were collected individually, considering each piece of the carcass that was studied, as follows: whole carcass, carcass remainder, whole viscera, abdominal fat, back, blood, breast, caeca, comb, drumsticks, feathers, giblet, gizzard, head, heart, intestine, liver, lungs, neck, ovary, pancreas, pelvis, proventriculus, rear, ribs, shanks, skeletal, skin, spleen, testes, thighs, thymus, trunk, wattles, and wings. Once the document evaluation was performed, thirty-nine explanatory variables were considered in the statistical analysis as follows: sex, slaughter age, carcass/piece weight, carcass yield, cold weight, muscle fiber density, muscle fiber diameter, pH, pH 24 h post mortem, pH 72 h post mortem, L* meat, a* meat, b* meat, L* meat 72 h post mortem, a* meat 72 h post mortem, b* meat 72 h post mortem, L* skin, a* skin, b* skin, drip loss, water-holding capacity, cooking loss, firmness, total work, shear force, hardness, springiness, cohesiveness, gumminess, chewiness, IMP, AMP, inosine, moisture, protein, fat, ash, collagen, and cholesterol. The methodologies used for the determination of each particular explanatory variable in each specific document were not registered, given the techniques and procedures followed for measurement collection were standardized as to be considered in research procedures. Additionally, this decision was made on the basis that when standardized techniques are used, even if differences across methods and procedures may exist, these are negligible, as supported by scientific evidence. The breed of the bird was used as classification criteria to determine the variability in carcass and meat quality traits across native breeds to elaborate a data-mining chi-squared automatic interaction detection (CHAID) decision tree.

### 2.2. Data Analysis

#### 2.2.1. Multicollinearity Preliminary Testing

Multicollinearity analyses were run prior to statistical analyses per se to ensure independence and discard strong linear relationship across predictors. In this way, noise or redundancy problems in the variables used were detected before data manipulation. The exclusion of unnecessary variables through multicollinearity analysis ensures that redundancies do not overinflate the variance explanatory potential [132]. The variance inflation factor (VIF) is used as a multicollinearity indicator, and it can be calculated by the use of the following formula:VIF = 1/(1 − R^2^),(1)
where R^2^ is the coefficient of determination of the regression equation.

A recommended maximum VIF value of 5 can be found in the literature [133]. Tolerance (1 − R^2^) is defined as the amount of variability in a certain independent variable that is not explained by the rest, and a value higher than 0.20 is recommended [134]. The multicollinearity statistics routine of the describing data package of XLSTAT software (Addinsoft Pearson Edition 2014, Addinsoft, Paris, France) was used to perform the multicollinearity test.

#### 2.2.2. Data-Mining CHAID Decision Tree

Classification, prediction, interpretation, and manipulation of discrete categorized data were performed using the chi-squared automatic interaction detection (CHAID) decision tree (DT). For this, the tree routine of the analyzing data package of the XLSTAT package (Addinsoft Pearson Edition 2014, Addinsoft, Paris, France) was used. Each internal node was built in the tree around a meat quality trait (input variables), while in the so-called pre-pruning process, a significance split criterion of the chi-square test was met (*p* < 0.05).

Pre-pruning or post-pruning methods avoid oversizing trees to prevent failure to seek the addition of features (branches) that add significantly to the overall fit [8]. As a result, a tree that exhaustively represents the significant relationships between independent variables is one from which those nodes that do not contribute to the overall prediction have been discarded. Moreover, CHAID additionally penalizes the complexity of the model. In this sense, the significant adjustment of the Bonferroni inequality by significance levels was used.

Chi-square tests were used to determine the configuration of the tree-building process. Each branch represents a test result (in a number of two or more), and each terminal node (or leaf node) represents a category level of the target variable (breed). The root node of the tree is the one at the top. Decisions are made at each node, and each data record continues through the tree along a path until the record reaches a leaf or terminal node in the tree [135].

Subsequently, cross-validation was performed to validate the set of predictors considered by measuring the differences between the prediction error for the tree applied to a new sample and a training sample. Cross-validation of the decision tree was performed using the ‘complexity parameter’ and cross-validation error to estimate how accurately the model generalizes for unseen data, that is, how well it performs or predicts. A ten-fold cross-validation was performed by keeping each sample record in the training sample and the study data. The resubstitution error rate measures the proportion of original observations that were misclassified by various subsets of the original tree. The goal is to determine the shortest tree that collects the largest number of significant relationships. However, the lowest replacement rate is not always the optimal option, as this tree will be biased. In the same way, large trees will put random variation in the predictions as they go past the outliers. For these reasons, instead of selecting a tree based on the resubstitution error rate, a ten-fold cross-validation is used to obtain a cross-validated error rate from which the optimal tree is selected. Ten-fold cross-validation involves creating X-random subsets of the original data, reserving one segment as a test set, building a tree for the remaining X − 1 segments, and evaluating the tree using the test segment. This is repeated for all cuts, and an estimation of the error is evaluated. The sum of the error in segments X represents the cross-validation error rate. The tree that produces the lowest cross-validation error rate is selected as the tree that best fits the data.

## 3. Results

### 3.1. Study Georeferencing

Figure 1 shows the distribution of studies across countries, with China (16 studies), Thailand (11 studies), and India (10 studies), all belonging to the Asian continent, being the most active countries in terms of research publications based on the study of local populations. On the European continent, Spain (nine studies) and Italy (seven studies) stand out in terms of the number of documents found.

### 3.2. Analysis of Model Reliability

Table 2 presents a summary of the values of tolerance and VIF for those variables for which VIF < 5 was reported. In this way, pH 72 h post mortem, L* meat 72 h post mortem, a* meat 72 h post mortem, sex, firmness, and chewiness were the variables discarded from further analyses.

### 3.3. Data-Mining CHAID Decision Tree

The data-mining CHAID decision tree obtained from the chi-square dissimilarity matrix is depicted in Appendix A. Chi-square-based branch and node distribution suggested observations significantly (*p* < 0.05) differed across native breeds. For a better understanding of the decision tree, the frequency of each breed at the different nodes and the tree structure are reported in Appendix A, respectively. In addition, Figure 2 summarizes the first branches of the decision tree. The first classification depended on the shear force values, and four groups were depicted (9.31–18.25; 18.25–34.58; 34.58–56; 56–102.97). After this classification, the observations were sorted into subgroups according to muscle fiber diameter, carcass/piece weight, and pH traits.

The robustness and validity of the results are supported by cross-validation tests performed afterward. In this regard, the number of erroneously classified observations for each genotype was computed. Figure 3 reports the number of observations that could presumably be ascribed to the different studied genotypes (native breeds) and enabled the development of a tool that gathers similarities/dissimilarities across the different worldwide genotypes as a method of inferring the degree of meat quality parameters introgression.

## 4. Discussion

The development of knowledge of local breeds is strongly limited by the availability of animals and infrastructures where such research attempts are carried out. This is due to the inequality in budgets that are allocated to local breeds as opposed to commercial hybrid strains [28,136]. The primary sector must demand support from the institutions to continue advancing the scientific knowledge related to local breeds. It should be noted that in a wide range of countries, such as those belonging to South America, where native breeds play an important role in livestock farming in alternative and backyard systems [137,138], only one study, whose main objective was the characterization of the carcass and chicken meat, could be found, and it was published in a non-indexed journal; hence, its impact in the research community may be consequently limited [114]. On the other hand, large multinational poultry integrators present their headquarters in countries such as the United States or China, from where markets historically developed, and which still hold a strong connection to commercial hybrid strains and other highly productive foreign breeds, which translates into greater scientific support provided to these genotypes [139].

Regarding the traits used in this research, pH 72 h post mortem, L* meat 72 h post mortem, and a* meat 72 h post mortem traits were discarded from further analyses due to multicollinearity problems present within the same traits when measured at a shorter period from the moment of slaughter. pH indicates the rate and the intensity of muscle acidification during post mortem time. In order to estimate the intensity of acidification decrease, many authors have measured pH at 24 and 72 h post mortem [126,140]. In the measurement of pH, some authors performed color measurements up to 72 h post mortem too [141]. However, these multicollinearity problems with measurements taken at 72 h post mortem indicate a lack of their representativity when prior sampling moments had been considered, thus suggesting the discarding of such variables from experimental models for meat characterization studies on local poultry breeds.

Additionally, sex was a redundant variable too. This may be ascribed to the fact that significant differences could only be found in the carcass/piece weight and yield variables between the different sexes. The lower selective pressure on production of local breeds when compared with broilers reduces the likelihood of differences in the meat quality of the males when compared to that of females, as was reported in commercial hybrid lines [142].

These multicollinearity problems were also reported for chewiness and firmness variables. On the one hand, chewiness can be defined as the product of hardness x springiness x cohesiveness [143]. Multicollinearity problems may arise due to the relationship of these parameters with the rest of the variables used to characterize texture. On the other hand, while firmness was defined as the peak force exerted when a sample was compressed to a depth of 1.5 cm, using a block of timber of identical dimensions to the sample, hardness was defined as the peak force exerted when a metal probe (flat cross-section, 10 mm in diameter) was driven into the sample to a depth of 1.5 cm [144]. In this way, these two variables may seem redundant when characterizing meat from the local chicken breeds, since they correlate highly.

Once all the redundant variables were eliminated, the decision tree was built. Figure 2 suggests the best discriminating abilities were reported for shear force. Shear force is indicative of the toughness of meat and conforms to sensory traits [145]. Indeed, some authors have suggested that muscle fibers from fast-growing chicken breeds have larger diameters than those of slow-growing chicken genotypes. Larger fiber diameters are often associated with meat toughening, and therefore, higher shear force [113,146].

The fact that shear force was reported to have a high explanatory potential is due to the great variability found in this variable across all the genotypes of local chickens across the world. In this regard, black-feathered Taiwan native chicken presented the highest shear force values, hence the total differentiation of this genotype in node 5. Such differentiation may be ascribed to the fact that although broilers’ growth is slower in this genotype than the rest, the rearing period (16–20 weeks) is short when compared with most local genotypes [79]. As a result, the black-feathered Taiwan chicken could act as a good control group in any study due to its considerably increased shear force values when compared to the rest.

A second ramification of the decision tree occurred, and samples from genotypes with lower shear force values were further divided depending on muscle fiber diameter. Some factors, such as genotype, body weight, sex, or age, have been shown to affect muscle-type fiber and diameter, and in turn, meat tenderness [147]. A positive relationship was described between muscle fiber diameter and meat toughness [148]. However, the relationship between fiber diameter and shear force has been reported to be variable when adjusted for age. Hence, slaughter age may be more decisive than fiber diameter in meat tenderness, since individuals’ age has a high influence on fiber type, as intermediate fibers change to white fibers [149]. High values of muscle fiber diameter can be found in genotypes with low shear force values but high slaughter ages. In this regard, Greenleg Partridge and Aseel breeds separated in nodes 7 and 8, respectively. In the literature, references to an advanced slaughter age of these two genotypes can be found, reaching up to 56 and 60 weeks for Greenleg Partridge and Aseel, respectively [57,66]. Higher testosterone levels have been shown to produce an increase in muscle fiber size [150,151]. Therefore, puberty in chickens can be expected to produce an inflection point in the growth of their muscle fibers, which may explain the aforementioned outputs.

A high discriminant power was also reported for carcass/pieces weight. The weight of different meat pieces and carcass weights result from complex interactions between genotype and environment where individuals are raised, and such interactions make this trait highly variable [152]. Some poultry breeds have been exposed to high selective pressure in terms of body size. Indeed, quantitative trait locus, with main effects on the shank length, growth, weight gain, and body and carcass weights have been mapped a priori, which may be indicative of the relevance that these traits have [92,153].

Chicken breast and thigh yields change during different stages of growth, since the dressing percentage steadily increases during this process [154]. A loss in carcass weight is produced when carcasses are introduced into cooling chambers due to the action of the forced circulation of cold air. This is why cold canal weight and carcass/pieces yield are two very frequent variables to be considered in carcass characterization studies.

pH also reported a strong discriminating ability across genotypes. Regarding other meat physical properties, pH is an important parameter, as it may directly affect other quality traits, including meat texture, color, and flavor [155].

High values of pH increase the ability of meat to bind water, since the meat fluid is bound by the protein and causes a harder meat texture [156]. Texture or tenderness have been identified as the most important factors that determine consumer satisfaction [157]. The relationship between texture and pH is controversial. While some authors indicate a linear dependence between these two parameters, others support the hypothesis that described a curvilinear dependence with the highest level for meat texture with pH values in the range of 5.8 to 6.3. The different proteolytic activity that leads to less tenderness during aging explains this fact. Thus, the increased texture found when the pH increases from 6 to 7 is attributed to increased calpain activity, which is maximized at neutral pH [158,159,160].

By contrast, the texture increase when pH falls below 6 is attributed to a higher effect of the acid protease. The reduction in sarcomere length has been suggested to be a major cause of increased meat toughness, and sarcomeres appear to increase in length as the final pH falls below 6.2. Furthermore, the direct activity of calcium ions on myofibrillar proteins is attributed to a decrease in texture, and this process is pH dependent. Meat toughness immediately post mortem is the same whether the meat is of a low, medium, or high pH when ZnCl_2_ inhibits aging [157,160].

Regarding meat color, chicken meat is translucent. When the light scattering is weak, since tissues have high pH values, the light path through the tissue will be relatively long, and the selective absorbance of light myoglobin and its derivatives will increase. By contrast, the light path through the tissue is relatively short, and the selective absorbance decrease when the light scattering is strong is due to low pH values. Therefore, all aspects of meat colorimetry are influenced by the translucency of the samples. Most researchers use non-specific colorimeters that have been designed to measure painted, metal, or plastic surfaces but are really unaware of the optical problems created by the translucency of chicken meat [161].

The pH of the meat directly influences the development of flavors in the Maillard reaction too. A pH between 4.5 and 6.5 favors the formation of nitrogenous compounds that add flavor to food [162,163]. Post-mortem aging causes the generation of many chemical-flavor compounds, including organic acids, free amino acids, sugars, peptides, and metabolites of adenine nucleotide metabolism that determine the final flavor of meat [164].

The difference between genotype prior and posterior classification in the present research and depicted in Figure 3 helps detect the phenotypical similarities/dissimilarities across local breeds with respect to carcass and meat traits. This may be better understood, for example, through the case of the Aseel breed. The Aseel breed posterior classification differs, with individuals belonging to this breed being ascribed to many other genotypes. The fact that this breed has been extensively studied, with nine investigations in two different countries, seeking to meet the needs of consumers in each area, being handled under different conditions and with a wide range of slaughter ages (between 5 and 60 weeks), may presumably make the variability in the quality of the carcass and meat of this breed wide. However, on many occasions, the attempts of producers of the breed to make it competitive against other breeds may translate into a cross of it with other breeds, which in turn may make it phenotypically similar to other breeds [36,44,50,54,57,110,111,116,123].

## 5. Conclusions

The present research can be used as a guide for the comprehensive evaluation of literature resources when determining the experimental design in studies approaching the characterization of carcass quality in local chicken genotypes. Results evidenced high variability in carcass and meat quality when local chicken breeds around the world were compared. Preliminary multicollinearity analyses suggested that pH and color measurements at 72 h post mortem can be avoided, since the information they offer can be supplemented with the rest of the variables collected at other moments after slaughter. Shear force presented the highest explanatory potential in the CHAID decision tree, which may be ascribed to the high diversity in the growth rate of the different genotypes studied. Muscle fiber diameter, carcass/pieces weight, and pH offer a wide range of information about carcass quality and are easily measurable parameters in the daily operation of a slaughterhouse (either directly or through sampling for a posteriori processing); hence, their consideration is strongly recommended. Additionally, the present statistical tool enables one to determine the suitability of animals belonging to certain breeds to conform to research control groups. This permits one to tailor studies rather efficiently, as the specific characteristics of the control group (to which the rest of the groups are compared) may better fit the rationale being studied. Last but not least, this research enables the selection of specific genotypes for their increased suitability for the production of particular meat pieces, with an increased acceptance within the diverse market niches but also to adapt those, which do not meet the need of consumers, which, in turn, may ensure the sustainability of the valuable local genetic resources that can be found worldwide.

## Figures and Tables

**Figure 1 foods-11-01700-f001:**
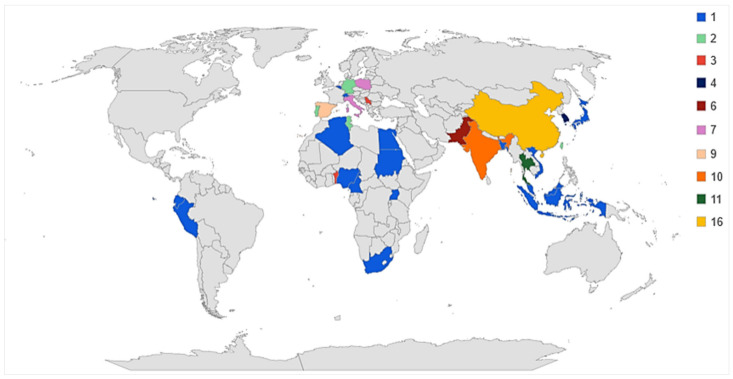
Territorial distribution and number of papers per country.

**Figure 2 foods-11-01700-f002:**
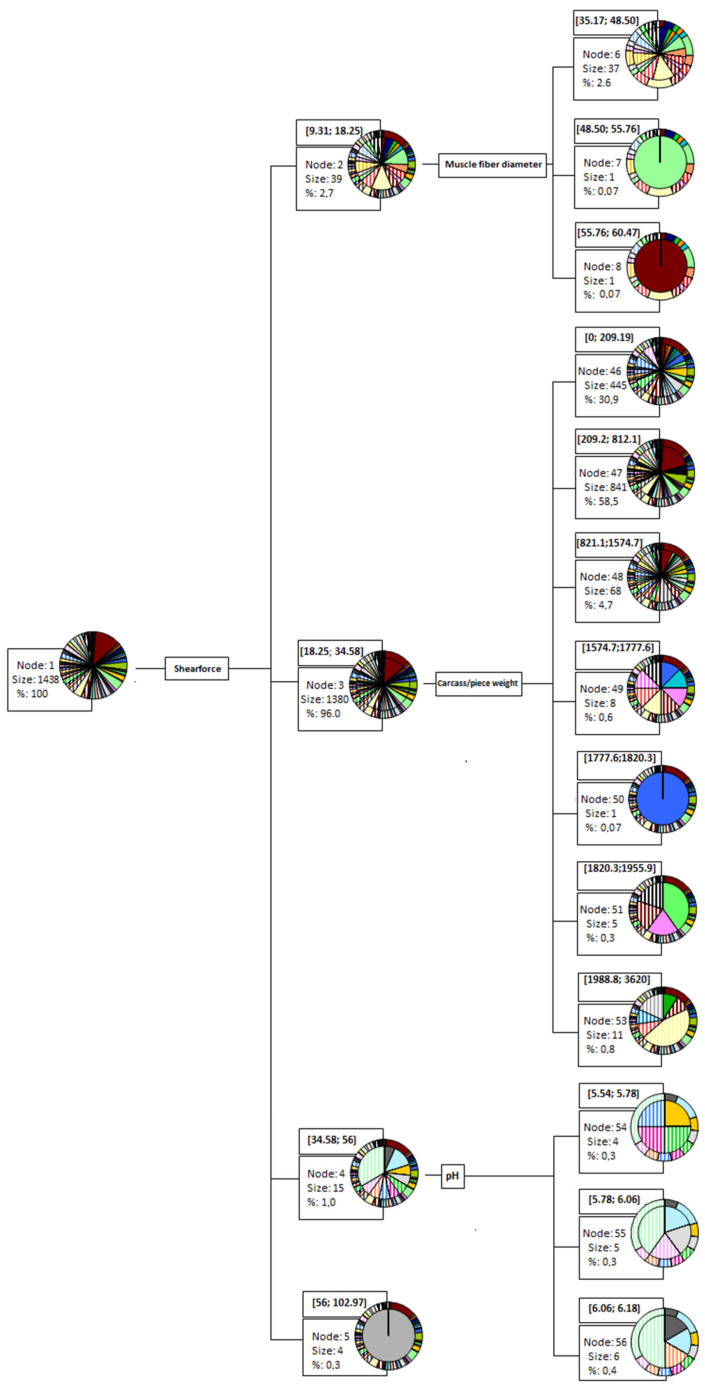
Graphic depiction of the most representative branches of the CHAID decision tree considering native breed genotypes as the clustering criterion.

**Figure 3 foods-11-01700-f003:**
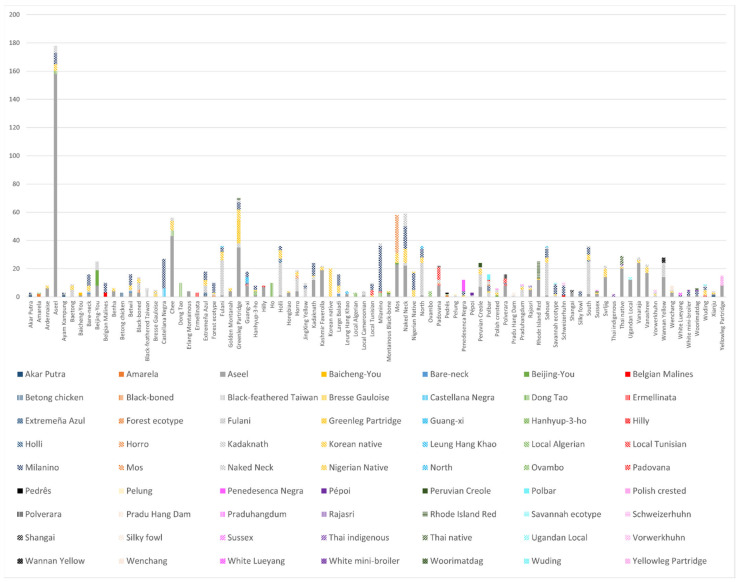
Graphical depiction of the prior and posterior classification of observations depending on their genotype.

**Table 1 foods-11-01700-t001:** Clusters, references, and units of the traits considered in the study.

Cluster	References	Trait	Units
Weight-related traits	[21,34,35,36,37,38,39,40,41,42,43,44,45,46,47,48,49,50,51,52,53,54,55,56,57,58,59,60,61,62,63,64,65,66,67,68,69,70,71,72,73,74,75,76,77,78,79,80,81,82,83,84,85,86,87,88,89,90,91,92,93,94,95,96,97,98,99,100,101,102,103,104,105,106,107,108,109,110,111,112,113,114]	Carcass/piece weight (g)	g
		Carcass yield (%)	%
		Cold weight	g
Histological properties	[39,44,60,69,77,108,109,115]	Muscle fiber density	Fibers/mm^2^
		Muscle fiber diameter	μm
pH	[21,34,35,36,37,38,40,42,43,44,45,46,49,51,54,56,57,61,62,63,65,66,67,68,69,71,72,74,75,76,77,78,79,80,86,88,91,93,97,102,103,105,107,108,109,116,117,118,119,120,121,122,123,124,125,126,127,128,129]	pH	
		pH 24 h post mortem	
		pH 72 h post mortem	
Color-related traits	[34,36,37,38,40,42,43,44,45,46,48,49,51,54,56,57,61,62,63,65,66,67,68,69,72,74,75,77,78,79,80,84,86,88,97,102,103,105,107,108,117,118,120,121,124,125,126,127,129]	L* meat	
		a* meat	
		b* meat	
		L* meat 72 h post mortem	
		a* meat 72 h post mortem	
		b* meat 72 h post mortem	
		L* skin	
		a* skin	
		b* skin	
Water-holding capacity	[34,36,37,40,42,44,45,46,48,49,56,57,60,61,62,63,65,66,67,69,71,72,74,75,76,77,78,79,80,85,86,88,93,97,107,108,109,112,113,117,118,121,122,124,125,126,127,128,129,130]	Drip loss	%
		Water-holding capacity	%
		Cooking loss	%
Texture-related traits	[34,37,38,40,42,43,44,45,46,48,49,54,56,57,60,61,62,63,66,67,69,71,74,79,80,85,86,88,97,107,108,109,113,117,118,120,121,122,124,125,126,127,128,129]	Firmness	kg s^−1^
		Total work	kg mm
		Shear force	N
		Hardness	N
		Springiness	mm
		Cohesiveness	N
		Gumminess	N
		Chewiness	kg mm
Content of flavor-related nucleotides	[45,52,60,81,113,121,126]	IMP	mg/g
		AMP	mg/100 g
		Inosine	mg/100 g
Gross nutrients	[20,21,35,38,40,42,43,44,45,46,47,51,53,56,59,60,62,63,64,65,68,69,71,72,74,76,79,81,82,84,87,88,91,93,96,97,103,104,106,107,108,109,111,113,116,117,118,119,121,122,124,127,128,129,130,131]	Moisture	%
		Protein	%
		Fat	%
		Ash	%
		Collagen	%
		Cholesterol	mg/100 g

**Table 2 foods-11-01700-t002:** Multicollinearity analysis of meat- and carcass-quality-related traits.

Statistics/Traits	Tolerance (1 − R^2^)	VIF ^1^
Chewiness	0.2468	4.0515
Gumminess	0.3126	3.1989
Hardness	0.4300	2.3258
Shear force	0.4867	2.0546
a* meat	0.5302	1.8862
b* skin	0.5635	1.7745
a* skin	0.5867	1.7044
Muscle fiber diameter	0.6164	1.6223
Cooking loss	0.6172	1.6202
L* skin	0.6191	1.6152
L* meat	0.6285	1.5910
Water-holding capacity	0.6418	1.5580
pH	0.7088	1.4108
Drip loss	0.7201	1.3886
pH 24 h post mortem	0.7415	1.3486
Moisture	0.7428	1.3462
b* meat	0.7458	1.3408
Total work	0.7875	1.2699
IMP	0.7978	1.2534
Springiness	0.8208	1.2183
Cholesterol	0.8264	1.2101
Cohesiveness	0.8981	1.1135
Collagen	0.8985	1.1130
Inosine	0.9044	1.1058
Carcass/piece weight	0.9133	1.0949
Carcass yield	0.9176	1.0898
Protein	0.9293	1.0761
AMP	0.9315	1.0735
Ash	0.9558	1.0463
Muscle fiber density	0.9692	1.0317
Cold canal weight	0.9732	1.0275
Average age	0.9740	1.0267
Fat	0.9792	1.0213

^1^ Interpretation thumb rule: VIF ≥ 5 (highly correlated); 1 < VIF < 5 (moderately correlated); VIF = 1 (not correlated).

## Data Availability

All data stemming from the present research are enclosed in the tables or as Appendix A. Any additional data will be made accessible from the corresponding authors upon reasonable request.

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
