# Peer review of "Variability of Meat and Carcass Quality from Worldwide Native Chicken Breeds"

_foods, 2022, doi:10.3390/foods11121700_

Round 1

Reviewer 1 Report

CHAID data mining as a method to study the variability of meat and carcass quality from worldwide native chicken breeds

This manuscript has results interesting to analyze the variability of meat and carcass quality about chicken breeds. I have little recommends.

Title

The word “CHAID” can not use in title because is a methodology for data anlysis as other statistic methods. I recommend this title: Variability of meat and carcass quality from worldwide native chicken breeds. This recommendation is about manuscript objectives.

Abstract

Without comments.

Introduction

The Figure 1 is not necessary because you can write in text about information.

Material and methods

The data analysis process was adequate to analyse variables. I have not comments.  

Results and discussion

These sections are adequate to present the analysis.  

Conclusion

Correct

Author Response

Reviewer 1

Comments and Suggestions for Authors

This manuscript has results interesting to analyze the variability of meat and carcass quality about chicken breeds. I have little recommends.

Response: We thank the reviewer for his/her kind comments.

Title: The word “CHAID” can not use in title because is a methodology for data anlysis as other statistic methods. I recommend this title: Variability of meat and carcass quality from worldwide native chicken breeds. This recommendation is about manuscript objectives.

Response: The title has been changed as proposed by reviewer 1.

Abstract: Without comments.

Introduction: The Figure 1 is not necessary because you can write in text about information.

Response: Figure 1 has been removed.

Material and methods: The data analysis process was adequate to analyse variables. I have not comments. 

Results and discussion: These sections are adequate to present the analysis. 

Conclusion: Correct

Response: We thank the reviewer again for his/her kind comments.

Reviewer 2 Report

In the present paper I carefully reviewed, the Authors aimed to investigate the differential clustering patterns of carcass and meat quality traits in local chicken breeds from around the world and to develop a method to productively characterize minority bird populations. Therefore, a meta-analysis of all the investigations dealing with native chicken breeds during the last 20 years was carried out by the Authors.

I would like to congratulate Authors for the good-quality of their article, the literature reported used to write the paper, and for the clear and appropriate structure.

The manuscript is well written, presented and discussed, and understandable to a specialist readership.

In general, the organization and the structure of the article are satisfactory and in agreement with the journal instructions for authors. The subject is adequate with the overall journal scope.

The work shows a conscientious study in which a very exhaustive discussion of the literature available has been carried out.

The Introduction section provides sufficient background, and the other sections include results clearly presented and analyzed exhaustively.

However, as specific comments, with the aim to further improve the quality of the paper, I suggesti to:

- The Introduction section could be further improved by adding a couple of sentences referring to recently published papers.

- The Conclusion section could be further improved;

- The Authors have to check if alle references have been cited in the text.

Author Response

Reviewer 2

Comments and Suggestions for Authors

In the present paper I carefully reviewed, the Authors aimed to investigate the differential clustering patterns of carcass and meat quality traits in local chicken breeds from around the world and to develop a method to productively characterize minority bird populations. Therefore, a meta-analysis of all the investigations dealing with native chicken breeds during the last 20 years was carried out by the Authors.

I would like to congratulate Authors for the good-quality of their article, the literature reported used to write the paper, and for the clear and appropriate structure.

The manuscript is well written, presented and discussed, and understandable to a specialist readership.

In general, the organization and the structure of the article are satisfactory and in agreement with the journal instructions for authors. The subject is adequate with the overall journal scope.

The work shows a conscientious study in which a very exhaustive discussion of the literature available has been carried out.

The Introduction section provides sufficient background, and the other sections include results clearly presented and analyzed exhaustively.

Response: We thank the reviewer for his/her kind comments.

However, as specific comments, with the aim to further improve the quality of the paper, I suggesti to:

- The Introduction section could be further improved by adding a couple of sentences referring to recently published papers.

Response: Added.

- The Conclusion section could be further improved;

Response: This section has been improved.

- The Authors have to check if alle references have been cited in the text.

Response: All the references have been checked to be cited in the text.